# Drinking to Cope Mediates the Association between Dyadic Conflict and Drinking Behavior: A Study of Romantic Couples during the COVID-19 Pandemic

**DOI:** 10.3390/ijerph20146332

**Published:** 2023-07-10

**Authors:** Amanda E. F. Hagen, Lindsey M. Rodriguez, Clayton Neighbors, Raquel Nogueira-Arjona, Simon B. Sherry, Laura Lambe, S. Hélène Deacon, Sandra Meier, Allan Abbass, Sherry H. Stewart

**Affiliations:** 1Department of Psychology & Neuroscience, Dalhousie University, Halifax, NS B3H 4R2, Canada; simon.sherry@dal.ca (S.B.S.); laura.lambe@dal.ca (L.L.); helene.deacon@dal.ca (S.H.D.); 2Department of Psychology, University of Florida, Gainesville, FL 33701, USA; lindsey.rodriguez1@gmail.com; 3Department of Psychology, University of Houston, Houston, TX 77204, USA; cneighbors@uh.edu; 4School of Psychology, University of Sussex, Brighton BN1 9RH, UK; r.nogueira-arjona@sussex.ac.uk; 5Department of Psychiatry, Dalhousie University, Halifax, NS B3H 2E2, Canada; sandra.meier@iwk.nshealth.ca (S.M.); allan.abbass@dal.ca (A.A.)

**Keywords:** COVID-19, alcohol use, couples’ conflict, drinking motives

## Abstract

The COVID-19 pandemic spurred public health measures to reduce viral spread. Concurrently, increases in alcohol consumption and conflict in romantic partnerships were observed. Pre-pandemic research demonstrated a bidirectional association between couples’ conflict and drinking. Recent research shows one’s drinking motives (proximal predictors of drinking behavior) can influence another person’s drinking in close relationships. It is possible that individuals are drinking to cope with distress following romantic conflict. The current study examined 348 cohabitating couples during the first lockdown in the spring of 2020. Our analyses examined coping motives as a mediator between dyadic conflict and drinking behavior using actor–partner interdependence models. Results showed that conflict was associated with greater reports of own drinking in gendered (distinguishable) and nongendered (indistinguishable) analyses through coping motives. Further, in mixed-gender couples, men partners’ coping motives predicted less drinking in women, while women partners’ coping motives predicted marginally more drinking in men. Partner effects may have been observed due to the increased romantic partner influence during the COVID-19 lockdown. While these results suggest that men’s coping motives may be protective against women’s drinking, more concerning possibilities are discussed. The importance of considering dyadic influences on drinking is highlighted; clinical and policy implications are identified.

## 1. Introduction

In the spring of 2020, businesses, travel, workplaces, and schools were closed across Canada and many jurisdictions around the world in response to the declaration of the COVID-19 virus as a pandemic [1]. As such, many cohabitating romantic couples found themselves spending extended periods of time in the home together. Couples also had highly limited opportunities to socialize with others or participate in activities outside the home. Romantic couples also faced additional pandemic-specific stressors such as shifts to working from home, mandatory homeschooling, and financial and/or health-related concerns.

The effects of the COVID-19 pandemic and associated public health measures have resulted in deleterious consequences to *individuals’* well-being, including increases in anxiety and depressive symptoms, increased loneliness, and decreased positive emotions and life satisfaction [2,3]. Further, some evidence suggested increased individual alcohol use and alcohol-related problems during COVID-19 lockdowns [4,5], particularly among those with higher levels of pandemic-related distress [6,7]. The population’s increase in alcohol use is of great public health importance due to the range of adverse consequences (e.g., injury, medical illness, suicide, and violence) associated with heavier drinking [8]. Despite increases in reports of stressful life events frequently associated with increases in romantic conflict (e.g., job loss and health changes) [9], there is less information available on the psychological impact of the pandemic on romantic *couples.* There is a particular dearth of studies using dyadic methods where the perspectives of both couple members are included.

### 1.1. Romantic Partnerships and Conflict in COVID

Romantic relationships are typically the largest source of social interaction and one of the most influential social relationships [10,11]. Indeed, romantic relationship satisfaction is a robust predictor of overall life satisfaction [12]. Research has begun to identify the impacts of the pandemic on romantic relationship functioning including effects on couples’ well-being and relationship satisfaction. For instance, married individuals experienced a steeper decline in well-being than unmarried people during the COVID-19 outbreak [13].

The COVID-19 pandemic and associated stressors (e.g., mandatory homeschooling) have increased ‘couples’ conflict’ [14]. This dyadic construct includes a variety of negative communication behaviors that arise out of disagreement or incompatibility including interruptions, contempt, rejection, condemnation, and communication avoidance [15]. At the extremes of conflict, reports of intimate partner violence roughly doubled in frequency and severity during the pandemic [16]. Moreover, couples who reported more COVID-19-related stress also reported more conflict and lower relationship satisfaction [17]. Increases in romantic conflict resulted in decreased intimate sexual behavior within the partnership during COVID-19 [18], further undermining relationship quality. The importance of examining romantic conflict during the COVID-19 pandemic is further underscored by the known deleterious effects of romantic conflict on life and relationship satisfaction over time [19,20,21].

### 1.2. Romantic Partnership Conflict and Drinking

A systematic review found the effects of drinking on marriages are overwhelmingly negative [22]. Indeed, heavy alcohol use and alcohol-related problems co-occur with marital conflict [22]. Other research suggests the relation between drinking and marital satisfaction is more nuanced. First, there are reported positive effects of matched substance use behavior in romantic couples, regardless of the quantity of use [23,24]. Additionally, a bidirectional relation exists between conflict and drinking in couples. Not only does drinking before a conflict result in more aggressive conflicts [25], but some research suggests conflict results in more alcohol use or alcohol-related problems [11,26]. For example, a daily diary study found that romantic conflict predicted increases in next-day drinking behavior [24]. This study also showed that the tendency to drink in response to earlier romantic conflict was stronger in women than in men [24].

Cooper’s drinking motives theory [27,28], describes how individuals may be motivated to drink to achieve different desired outcomes. These different motives are outlined in terms of the type of reinforcement desired (i.e., positive or negative reinforcement) and the source of reinforcement (i.e., internal or external to the individual). Crossing these two dimensions of desired outcomes yields four distinct drinking motives: social (positive-external motives; to increase social affiliation), enhancement (positive-internal motives; to heighten pleasurable sensations or affect), conformity (negative-external motives; to alleviate/avoid social rejection), and coping (negative-internal motives; to reduce undesirable internal states) [27]. Of these four motives, coping motives appear the riskiest or among the riskiest [29]; they consistently predict heavy drinking and alcohol-related problems both cross-sectionally [30,31] and longitudinally [28,32,33].

Coping motives provide one possible explanation for drinking behavior following conflict. Drinking to cope with interpersonal conflict (including romantic conflict) explains 36% of the variance in coping drinking motives [34]. This supports a coping-motivated link in the couples’ conflict-to-drinking relation. Additionally, an individual’s drinking motives (including their coping motives) may be influenced by the drinking motives of close others. This has been theorized to occur via social learning: while one cannot directly observe the motivations of others, an individual can witness their partner being reinforced or punished, which may lead them to drink to achieve similar outcomes [35]. While it is well established that romantic partners influence one another’s drinking behavior [36], it has only recently been established in pre-pandemic research that an individual’s drinking motives can influence their partner’s drinking behavior. Specifically, one partner’s coping motives were longitudinally associated with the other partner’s drinking behavior as mediated through the other partner’s coping motives [37]. This points to the importance of examining partner effects in this field.

Another pre-pandemic study from Lambe et al. examined the links between couples’ conflict and alcohol outcomes in a sample of 100 emerging adult couples [38]. They examined the individual’s own coping motives and the partner’s drinking motives as possible mediators of the conflict—drinking outcomes link [38]. In indistinguishable dyad analyses (i.e., not distinguished by gender), this study showed that the individual’s coping-with-depression motives mediated the relation between dyadic conflict (combined reporting by both partners) and the individual’s alcohol-related problems [38]. Within the mixed-gender couples, this mediation held true for women but not men. This pattern was consistent with prior gendered findings on conflict-related drinking [24]. Both the indistinguishable dyads and distinguishable dyads (by gender) analyses found actor effects but no partner effects. Thus, no evidence was found that the partner’s coping motives influenced the individual’s alcohol-related problems [38].

While there were important insights offered by Lambe et al.’s study [38], there were also limitations. First, this study recruited emerging adults, which may limit generalizability to longer-term romantic relationships since partner drinking influence has been shown to vary by stage of a relationship, increasing with marriage [23]. Further, with only 100 couples, their study may have been underpowered to detect smaller magnitude partner effects [39]. Additionally, Lambe et al. [38] examined alcohol-related problems as their outcome. Increased drinking levels are the more proximal outcome of drinking to cope, which may lead to alcohol-related problems over time [30]. While examining alcohol-related problems may have been an appropriate outcome given Lambe et al.’s [38] longitudinal design, it remains important to test their model with drinking behavior as the theoretically more proximal outcome, particularly given the adverse public health consequences of increased heavy drinking during the pandemic [5,40].

### 1.3. The Current Study

Given the increases in both couples’ conflict and drinking during the COVID-19 pandemic [4,5,17], it is important to test the influence of the drinking motives of both individuals in a couple in response to dyadic conflict on both couple members’ heavier drinking behavior. In other words, it is important to test both actor and partner effects. The current study was a conceptual replication of Lambe et al. [38] and an extension to a larger, older, community-recruited sample and to drinking behavior as the outcome, conducted in the context of stay-at-home orders during the COVID-19 pandemic. We utilized a cross-sectional design and the Actor–Partner Interdependence Model (APIM; [41]). We reasoned that results could help shed light on the consequences of the pandemic, along with the associated public health viral containment measures put in place in the spring of 2020, on drinking behavior in couples.

We hypothesized we would detect partner effects despite their absence in Lambe et al. [38] due to our larger sample size (partner effects tend to be smaller, e.g., [37]) and due to the probable greater influence of a cohabitating partner on an individual’s behavior during the stresses and isolation of lockdown. Additionally, the various unique stressors associated with lockdown (e.g., health, financial, homeschooling, and loss of leisure activities) could have increased conflict (e.g., negative partner behaviors and verbal aggression) and the influence of conflict on members of a couple. We reasoned that the increase in time spent with one’s romantic partner and corresponding decrease in time spent with other individuals outside the home may amplify the influence of a romantic partner on one’s own behavior. We specifically hypothesized that:

**H1:** 
*Dyadic conflict would have an indirect effect on increasing drinking behavior through one’s own coping motives (actor effects). This would hold true for both indistinguishable and distinguishable (by gender) dyads analyses [38].*


**H2:** 
*While both were hypothesized to be significant, we expected that mediation effects for the actor effects would be significantly larger in women than the same mediation for men in the distinguishable (by gender) dyads analysis [24,38].*


**H3:** 
*Dyadic conflict would have an indirect effect on increasing drinking through one’s partner’s coping motives (partner effects; [37]) in both indistinguishable and distinguishable dyads analyses.*


## 2. Materials and Methods

### 2.1. Participants

Participants were 758 romantic cohabitating couples in Canada (*N* = 1516 individuals). To meet study eligibility, Qualtrics panelists must have been living in Canada throughout April 2020, been at least 19 years of age, been involved in a romantic relationship with a partner who is also at least 19 years of age, cohabited with that same romantic partner throughout April 2020, reported that they and their partner were following COVID-19-related stay-at-home advisories in their local jurisdiction in April 2020, and confirmed that their romantic partner was also willing and available to participate. Essential workers were excluded as we wished to examine couples staying at home together during lockdown. Additional descriptive statistics can be found in four other published papers from this dataset [14,42,43,44]. This study was approved by a university research ethics board (#2020-5166).

Only couples in which both partners reported drinking on at least one occasion in April 2020 were included in the present analyses (*n* = 348 couples). Non-drinkers (couples where one or both members reported consuming alcohol zero times in the reporting period; *n* = 410 couples) were excluded due to the zero-inflated distribution created by these cases for the outcome variables and due to the absence of coping motives for non-drinkers (i.e., drinking motives are only queried of drinkers). Demographics of the final sample are provided in Table 1. Descriptive statistics for key variables can be found in Table 2 and Table 3 for the total sample and subsamples by gender, respectively. Notably, the current sample contains a higher proportion of “binge drinkers” (i.e., heavy episodic drinkers) in both the men and the women than pre-pandemic Canadian norms [45]. Additionally, men showed higher rates of binge drinking than women in our sample for both peak and typical drinking occasions, consistent with gender differences in binge drinking rates seen prior to the pandemic [45]. Chi-square analyses were conducted to compare those included in the analyses vs. those excluded for having a non-drinker as one or both members of a couple. Notably, the subsample used in the present analyses were more educated, reported higher income, and more full-time employment, than those excluded from analyses for not consuming alcohol during the period assessed. Additionally, there were significant differences in ethnicity: those excluded from analyses for being non-drinkers were disproportionately East Asian and South Asian. Chi-square analyses yielded no significant differences in relationship composition, relationship status, if living with children or not, other ethnic categories, gender, age, or relationship length between those that consumed alcohol (and thus were included in the present analyses) vs. those that were excluded from the analyses. Lastly, couples included in the analyses did not report significantly more romantic conflict, coping motives, peak alcohol consumption, or typical alcohol consumption quantity than those excluded; however, individuals in couples that were included in the analyses (in which both partners reported drinking) reported drinking at a significantly greater frequency than drinkers whose partner did not drink and thus were excluded from analyses (see Table 2).

### 2.2. Measures

#### 2.2.1. Demographics

Demographic and relationship variables were assessed such as gender, sex, age, racial identity, employment status, educational attainment, relationship status, relationship length, as well as adherence to stay-at-home advisories, essential worker status, and COVID-19 infection [42,43]. Age was controlled in the main analyses. Income was controlled in supplemental sensitivity analyses.

#### 2.2.2. Dyadic Conflict

Conflict was assessed using the 7-item Partner-Specific Rejecting Behaviors Scale [46]. Participants were asked to report on their behavior towards their partner (e.g., “I was angry or irritated with my partner”, “I snapped or yelled at my partner”) on a Likert scale from *1-strongly disagree* to *9-strongly agree*, in which higher sum scores (possible scale range 7–63) indicated more severe conflict behavior. Consistent with previous research [38,47], scores from each member were averaged into a dyadic conflict score for analysis. This score represented equal contributions from each partner and captured the fact that couples’ conflict is an inherently dyadic construct. Notably, this scale measures negative conflict behaviors (i.e., not just the presence of disagreement, but rather behaviors such as ignoring, insulting, and criticizing) but does not measure intimate partner violence or physical aggression. This scale has shown strong psychometric properties, i.e., internal consistency [47]. Additionally, each individual’s reports on their own conflict behavior toward their partner were highly associated within couples, *r* = 0.78, *p* < 0.001, providing psychometric support for the averaging of the two scores into a single dyadic conflict measure. Internal consistency for the present sample can be found in Table 4.

#### 2.2.3. Drinking to Cope

Drinking motives were measured using the two coping items from the Brief Alcohol Motive Measure (BAMM; [48]). Participants were asked to report how frequently during the month of April 2020 they drank for various reasons. Responses were made on a set of sliding scale visual analogue scales (VAS) anchored from *never* to *always.* Each VAS item was scored on a continuum from 0–100, divided by 10 for analyses for consistency with Bartel et al. [48]. Each item represented a distinct motive for drinking. This scale has been validated against the original multi-item Modified Drinking Motives Questionnaire-Revised [49] and has been shown to have similar levels of concurrent and predictive validity to the original measure in relation to drinking levels and alcohol-related problems [48]. The drinking to cope with anxiety and drinking to cope with depression items were averaged into a single coping motives scale [42] given their high correlation in the present sample (*r* = 0.82, *p* < 0.001) and to increase reliability of measurement (Table 1). 

#### 2.2.4. Drinking Behavior

The Quantity/Frequency/Peak Alcohol Use Index [50] was used to assess drinking behavior during the month of April 2020. Analyses used a composite of three measures: drinking frequency, i.e., how many days of consuming alcoholic beverages in the month, with 11 ordinal forced choice options ranging from “once a month” to “every day”, which were translated to days of drinking per month (e.g., “4 times per week” was coded as “16”); typical quantity, i.e., how many drinks were typically consumed on days with alcohol consumption (with options ranging from 0–25+; in which “25+” was coded as 25.5); and peak quantity, i.e., how many drinks were consumed on the single instance of heaviest consumption (with options ranging from 0–25+, with responses of “25+” coded as 25.5). Frequency was used to identify individuals who were non-drinkers during April 2020.

### 2.3. Procedure

Qualtrics Panels, a survey management service that recruits from a large pool of potential participants, was used to gather the current data based on researcher-specified criteria. A total of 3292 couples began the survey and were assessed on the eligibility criteria. Participants who were eligible and provided informed consent were asked questions about a variety of constructs relevant to the pandemic including their demographics, relationship conflict, coping motives for alcohol use, and drinking behavior. Their partner completed the same questions in relation to their own demographics, conflict behavior, and drinking motives and behavior. All items were completed in early July 2020, and retrospectively queried for the month of April 2020, during which all Canadian provinces were under stay-at-home advisories under Public Health Emergency acts [51].

Three mechanisms were in place to screen out invalid responses: a question assessing commitment to giving honest answers while completing the survey, filter questions for which participants were instructed to select a specific answer to determine attention and reading of instructions, and a speeder check to ensure adequate time was spent completing the survey. Of those recruited to participate, couples were excluded for the following reasons for at least one partner: partner unwilling to complete the survey or partial completion (*n* = 1996), failed attention/speeder check (*n* = 183), or not meeting specific eligibility criteria (*n* = 355). These exclusions yielded a sample of 758 couples. The current study also required that both members of the couple consumed alcohol at least once in April 2020, yielding our final sample (*N* = 348 couples). Participants were compensated through Qualtrics according to their guidelines.

## 3. Results

### 3.1. Analysis Plan

Given the non-normal distribution and count nature of the drinking variables, we elected to use generalized structural equation models specifying a negative binomial distribution for the outcome variable (drinking) due to the non-normal distribution of the data. The models were run using listwise deletion. No cases were removed from the main analyses due to missing data. Drinking was operationalized as a latent variable comprising drinking frequency, typical quantity, and peak quantity. APIM was used to estimate actor and partner effects of the mediator (coping) on the outcome (drinking) [39]. Romantic conflict was operationalized as a between-dyads variable as it was the composite of both partners’ reports as in Lambe et al. [38]. Data and syntax are available from the corresponding authors on request.

Evidence for mediation is established when the 95% confidence interval from bias-corrected bootstrapping with 1000 resamples does not contain the value 1. This is due to the confidence interval being estimated around the incidence rate ratio which surrounds 1 instead of 0. To be inclusive of all participants regardless of sexual orientation, we first tested the components of the path model and indirect effects of romantic conflict on drinking through coping motives in the entire sample in indistinguishable dyad analyses (i.e., where partners are not distinguished by gender). Then, using the mixed-gender couples only, we evaluated whether each path and subsequent indirect effects, differed by gender (i.e., whether each path was different for men and women) in distinguishable dyads analyses, with gender as the distinguishing feature.

Specifying a negative binomial distribution for the outcome variable results in the estimates representing log-linked coefficients. One way to return the estimates into an interpretable format is through their exponentiated form: incidence rate ratios (IRRs). IRRs surround the number one and represent a percent higher or lower in the outcome as a function of a one-unit increase in the predictor. For example, an IRR of 1.22 for the effect of conflict on drinking means that every one-unit more in conflict reported corresponds to 22% more drinking. Similarly, an IRR of 0.95 indicates that every one-unit more in conflict reported corresponds to 5% less drinking. See Table 5 and Table 6 for IRRs for the indistinguishable (non-gendered) and distinguishable (gendered) APIM models, respectively.

Demographic information was probed for significant associations with drinking behavior (outcome variable). Age and income were used as covariates as they had the strongest relationships with drinking behavior. Other variables that were associated with drinking behavior in our sample (e.g., relationship length, having children in the home or not, employment status, and education level) were associated less strongly with drinking behavior and very highly correlated with each other (e.g., age and relationship length; income and employment status). Thus, age and income were selected as covariates for parsimony in possible confounding variables. While age was covaried in the main analysis, income was added as a covariate in a sensitivity analysis due to the power lost when including income, as 6% of our sample opted to not report their income.

### 3.2. Descriptive Statistics

Descriptive statistics are found in Table 1, Table 2 and Table 3 and bivariate correlations are in Table 4. All outcomes were positively and significantly correlated with each other.

### 3.3. Indistinguishable Dyadic Analysis

Conventional fit indices are not calculated for generalized structural equation modelling; however, the significance of paths within the model suggested a strong fit. Standardized path coefficients are presented in Figure 1 and indirect effects and IRR values in Table 5.

Within the latent variable of drinking, frequency was constrained to load at *b* = 1.00, while peak quantity loaded at *b* = 0.62, 95%*CIs* [0.58, 0.66], *p* < 0.001 and typical quantity significantly loaded at *b* = 0.43, 95%*CIs* [0.39, 0.48], *p* < 0.001. Age was a significant covariate, *b* = 0.03, 95%*CIs* [0.03, 0.03], *SE* = 0.001, *p* < 0.001.

Dyadic conflict significantly predicted greater drinking to cope motives, *b* = 0.90, 95%*CIs* [0.77, 1.04], *SE* = 0.07, *p* < 0.001. A significant positive actor effect for coping motives to drinking behavior was observed, *b* = 0.12, 95%*CIs* [0.09, 0.16], *SE* = 0.02, *p* < 0.001. No partner effect for coping motives to drinking behavior was observed, *b* = −0.01, 95%*CIs* [−0.04, 0.03], *SE* = −0.33, *p* = 0.741. Additionally, dyadic conflict directly predicted greater drinking behavior even after accounting for coping motives, *b* = 0.08, 95%*CIs* [0.02, 0.15], *SE* = 0.03, *p* = 0.010. Both partners’ drinking behaviors were significantly and positively associated with each other, *b* = 0.55, 95%*CIs* [0.43, 0.67], *SE* = 0.06, *p* < 0.001 (Figure 1).

The direct, indirect, and total effects are in Table 5. The mediation model was significant for actors: one’s own coping motives positively mediated the relation between dyadic conflict and one’s own drinking behavior. The partner’s coping motives did not significantly mediate the relation between dyadic conflict and the individual’s drinking behavior. In other words, actor effects were detected but partner effects were not. Overall, the model resulted in an IRR value of 1.22 (Table 5).

### 3.4. Distinguishable Dyads Analysis

The significance of paths within the generalized structural equation model suggested a strong fit. Standardized path coefficients are in Figure 2 and indirect effects and IRR values in Table 6.

Within the latent variable of drinking, drinking frequency was constrained to load at *b* = 1.00 for both genders. For men, peak quantity loaded at *b* = 0.65, 95%*CIs* [0.60, 0.69], *SE* = 0.02, *p* < 0.001, and typical quantity significantly loaded at *b* = 0.44, 95%*CIs* [0.39, 0.49], *SE* = 0.02, *p* < 0.001. For women, peak quantity significantly loaded at *b* = 0.56, 95%*CIs* [0.51, 0.61], *SE* = 0.03, *p* < 0.001, and typical quantity significantly loaded at *b* = 0.35, 95%*CIs* [0.29, 0.41], *SE* = 0.03, *p* < 0.001. Age was a significant covariate for both men, *b* = 0.04, 95%*CIs* [0.03, 0.04], *SE* = 0.001, *p* < 0.001, and women, *b* = 0.03, 95%*CIs* [0.03, 0.03], *SE* = 0.002, *p* < 0.001.

Dyadic conflict significantly predicted greater drinking to cope motives for both men, *b* = 0.98, 95%*CIs* [0.81, 1.13], *SE* = 0.08, *p* < 0.001, and women, *b* = 0.87, 95%*CIs* [0.69, 1.06], *SE* = 0.09, *p* < 0.001. One’s own coping motives predicted higher levels of one’s own drinking behavior for both men, *b* = 0.08, 95%*CIs* [0.03, 0.12], *SE* = 0.02, *p* = 0.002, and women, *b* = 0.19, 95%*CIs* [0.14, 0.25], *SE* = 0.03, *p* < 0.001 (actor effects). Notably, the partner paths of coping motives on drinking (i.e., partner’s coping motives predicting own drinking behavior) were significant in predicting women’s drinking *b* = −0.06, 95%*CIs* [−0.12, −0.002], *SE* = 0.03, *p* = 0.041, and marginal for men’s drinking, *b* = 0.05, 95%*CIs* [−0.002, 0.11], *SE* = 0.03, *p* = 0.062. These associations were opposite in direction in men and women: women partners’ coping motives marginally predicted *greater* drinking by men, while men partners’ coping motives significantly predicted *less* drinking by women. Dyadic conflict did not significantly directly predict drinking behavior for men, *b* = 0.06, 95%*CIs* [−0.01, 0.12], *SE* = 0.03, *p* = 0.092, or women, *b* = 0.06, 95%*CIs* [−0.01, 0.14], *SE* = 0.04, *p* = 0.098, after accounting for coping motives.

The direct, indirect, and total effects are in Table 6 for men and women. Women’s own coping motives significantly mediated the relation between dyadic conflict and their own drinking behavior but the mediation pathway for their partner’s drinking behavior was only marginally significant. In other words, higher conflict was positively associated with women’s coping motives which in turn predicted higher rates of women’s drinking and marginally more drinking in their male partners. Men’s coping motives significantly mediated the relation between dyadic conflict and their own drinking behavior but significantly mediated the relation between dyadic conflict and their partner’s drinking behavior in the *inverse* of the expected direction. In other words, higher conflict was associated with more men’s coping motives which in turn predicted higher rates of men’s drinking but lower drinking in their women partners. 

Significant partner effects were found for women’s drinking and only marginal partner effects were found for men’s drinking. However, the difference between these two effects themselves was not significant as the 95% confidence intervals for the mediation pathways overlapped.

Gender differences in actor effects were established by comparing 95% confidence intervals for the mediation pathway of dyadic conflict and own drinking behavior through one’s own coping motives for women versus men (Table 6). The direction of the difference was consistent with H2, in which we hypothesized larger effect sizes predicting women’s drinking than men’s drinking. However, the 95% confidence intervals for IRR values (i.e., exponentiated slope value) for the mediation pathways did overlap for women and men. Thus, while both pathways were independently statistically significant, the effect was not significantly larger for women than for men.

### 3.5. Sensitivity Analysis

The above analyses were also run with income as a covariate. Notably, *n* = 24 couples opted to not report income, resulting in missing data, and those that did not report income were significantly different than those who did in several important ways. Those who did not report income, on average, reported drinking fewer beverages per drinking occasion, reported significantly less drinking to cope with anxiety and drinking to cope with depression motives, and were significantly older. Thus, controlling income may have introduced systemic error into the results, as well as slightly lowered the power of the analyses.

The sensitivity analyses largely replicated the main analyses. For the distinguishable dyad analyses, the significant direct effect from conflict to drinking behavior when controlling coping motives became marginal when controlling income (*b* = 0.06, *SE* = 0.03, *p* = 0.063). For the indistinguishable dyad analyses, one component of the mediation pathway from conflict to women’s drinking via men’s coping motives (i.e., from men’s coping motives to women’s drinking) became marginal when controlling income, *b* = −0.06, *SE* = 0.03, *p* = 0.054. All other effects remained as reported in the main analyses (see Appendix A).

## 4. Discussion

Our study examined the role of coping motives as a mediator between dyadic conflict and drinking behavior in cohabitating romantic couples during the COVID-19 lockdown while controlling for age. Overall, both distinguishable and indistinguishable dyadic analyses showed that coping motives mediated the association between conflict and drinking behavior, such that each one unit increase in conflict would predict a 26–27% increase in drinking behavior—a small but meaningful effect. The effect sizes found in these analyses are consistent with recent arguments that most psychological phenomena of research interest are complex, in that the causes for such phenomena are highly multifactorial in nature and each individual cause exerts small effects. Therefore, studying and reporting small effects contributes to a cumulative science [52].

### 4.1. Actor Effects

Consistent with H1 and pre-pandemic results by Lambe et al. [38], romantic conflict was associated with greater reports of one’s own drinking behavior, mediated through their own higher coping motives. We replicated this finding in both indistinguishable dyad analyses with the full sample and in distinguishable (by gender) analyses with mixed-gender couples. This mediation effect would be hypothesized by drinking motives theory [28], providing evidence that individuals in cohabitating romantic relationships may increase their drinking to cope with negative affect (depression and anxiety) following couples’ conflict. This mediation suggests that coping motives could be a mechanism by which conflict may lead to more drinking and possibly to later alcohol problems. It is important to note that coping-related drinking is a particularly risky drinking motive and is more strongly associated than other motives with alcohol-related problems [30,53]. Lastly, this finding may partially explain increases in drinking during the COVID-19 pandemic [4,5] as increases in couple’s conflict have been reported during lockdowns [17] and in relation to the stressors faced by couples during lockdown [14].

Additionally, conflict had a direct effect on drinking behavior that was not accounted for by (mediated through) coping drinking motives in the indistinguishable dyad analyses. This suggests there are other mechanisms not included in the model that led members of a couple to greater drinking behavior following conflict (e.g., drinking as a shared activity to repair conflict [24]; coping with other forms of negative affect such as anger or shame [54,55,56]). However, it is important to note that this direct effect was neither found in the distinguishable (by gender) analyses nor when controlling the indistinguishable dyad analyses for income (see Appendix A). This suggests that it may be a spurious finding—at least partially explained by variance in income—and that coping motives may indeed fully mediate the association between romantic conflict and drinking.

Actor effects (ways in which one’s own coping motives predicted one’s own drinking behavior) were significant for both genders. Contrary to H2, the mediation of conflict on drinking behavior by coping motives was not significantly stronger in women than men. While past research indicates women are more likely to modulate their drinking behavior in response to conflict [24,38], this was not supported by our analyses. However, some findings suggest these effects may be more likely for the quantity of alcohol consumed, rather than the frequency of drinking occasions [7,24,57]. The current analyses combined measures of alcohol use into a single latent construct, which may have obscured gender differences in the quantity of alcohol consumed in response to distress. Separating the various indices of alcohol use behavior in future studies may yield more consistent findings on gender differences in alcohol use in response to romantic conflict.

Overall, these findings replicate previous findings that coping motives mediated the association between romantic conflict and alcohol-related problems and extend the results into a sample across adulthood, particularly with middle and late adulthood represented. However, it is worth noting the ways in which the older age of our sample may impact the variables assessed. There are inconsistent findings on whether verbal aggression decreases with age [58,59], with more robust findings that physical aggression and intimate partner violence decrease with age [60]. Additionally, there are several studies that indicate that mechanisms of conflict (e.g., personality predictors) and consequences of conflict (e.g., impact on health, depressive symptoms) are consistent across the lifespan [61,62,63,64]. Age is significantly correlated with relationship length and thus both are captured by covarying for age in these analyses; it is possible that aggression decreases with age and/or that conflict stabilizes over decades of partnership. Nonetheless, while rates of alcohol use and intimate partner violence decrease with age, these remain significant public health problems in older adults [65].

### 4.2. Partner Effects

Our third hypothesis [H3] predicting significant and positive partner effects was not supported. Men reported marginally more drinking behavior associated with their partner’s coping motives but this effect did not reach statistical significance. The inverse pattern was true for women, in that women reported significantly *less* drinking associated with their partner’s coping motives. However, these two effects were not statistically significantly different from each other in magnitude and both likely reflected small effects. Effect sizes are typically smaller for partner effects than actor effects in dyadic analyses. Additionally, we did not find observable partner effects in the indistinguishable analyses.

We hypothesized an adverse partner effect for both men and women, in which one partner’s coping motives would predict the other’s drinking behavior, but the *protective* partner crossover effect observed for women’s drinking behavior was not anticipated. While there is robust evidence of partner influence on drinking behavior [36,66], there is emerging evidence that one’s own drinking behavior can be influenced by the drinking *motives* of important network members. This research has primarily focused on peers [67,68,69], but a handful of studies have examined and supported this effect in romantic couples [37]. However, the existing literature and the partner influence hypothesis would predict a consistent *positive* association between a partner’s coping drinking motives and an individual’s drinking behavior regardless of gender [66]. In contrast, our data show higher coping motives in men were associated with *less* women’s drinking behavior in mixed-gender couples. ‘Emotion work’ may be one explanation for this unanticipated effect. Emotion work is a broad concept that encompasses activities that enhance a partner’s emotional well-being and provide emotional support, which may also explain gender inequities in relationships in other domains [70]. While emotion work enhances marriage satisfaction [71], degree and inequity of emotion work are negatively associated with well-being, particularly for individuals married to a man [72,73]. Research and gender role theory converge to suggest that women in mixed-gender relationships tend to be more concerned about and in-tune with emotion work [74].

Thus, it is possible that married/cohabiting women abstain from increased drinking in response to the coping-motivated drinking exhibited by their men partners following romantic conflict to reserve internal resources for emotion work. Indeed, a similar effect has been noted with depressive symptoms during the pandemic: women with depressive symptoms are more likely to be concerned about the impact of their symptoms on their spouse and to perform emotion work to modulate that impact compared to men with depressive symptoms [75]. A similar compensatory effect has been reported regarding drinking, where men’s distress was associated with a decreased likelihood of binge drinking behavior in their women partners [76]. It is important to note that this pattern of decreased consumption in response to men’s drinking to cope may be protective for women’s drinking while simultaneously being risky for intimate partner violence. Indeed, one study found that the highest rates of intimate partner violence occurred in couples in which the husband was a heavy-drinker and the wife was not [77]. Research is required to directly test the proposed emotion work mechanism, to probe specific motivations by women to drink less in response to their husband exhibiting more coping-motivated drinking, and to examine the beneficial and harmful consequences of this pattern.

Contrary to H3, no significant partner-coping mediation between dyadic conflict and drinking behavior was found in either the indistinguishable dyadic analyses with the full sample or for women’s coping predicting men’s drinking in the distinguishable dyadic analyses. Nonetheless, the lower limit of the confidence interval for the partner effect on men’s drinking included but did not go past zero (or below one for the IRR values: i.e., a marginal effect). This finding begets further investigation, as it is possible that a larger sample or more at-risk sample may yield the hypothesized significant partner effects on men’s drinking. The absence of partner effects in the distinguishable dyads analysis may have been due to the gendered partner effects being of similar magnitude but in opposite directions. This may have resulted in the washing out of overall partner effects when collapsing across gender. This possible masking underscores the importance of our following up analyses in the full sample with gendered analyses.

These findings on partner effects also have theoretical implications. Both the partner influence hypothesis and most extant evidence suggests partner influence occurs via one’s own drinking motives (i.e., the partner’s motives influence one’s own motives, which then proximally predict one’s own drinking) and that influence occurs in the direction of risk (e.g., [37]). However, these results provide preliminary evidence of partner effects being protective. The possible, though currently untested, mechanism of emotion work may add to the existing theory. Additionally, while the current study did not probe the mechanism of individual motives influencing partner motives directly, the marginal effect for men’s drinking would be consistent with the partner influence hypothesis. If replicated, and if tested in a serial mediation model (i.e., conflict → partner A coping motives → partner B coping motives → partner B alcohol use), it would further strengthen the extant literature. Furthermore, most of the literature in this area samples emerging adult couples or youth; the present study provides preliminary evidence of drinking motives influencing the behavior of others in middle adulthood. It is possible that partner influence decreases in strength with age or life stage. Additionally, our larger sample may have been more adequately powered to detect partner effects than the Lambe et al. [38] study, though possibly not large enough to detect significant partner effects for men’s drinking, as partner effects overall tend to be small after controlling for actor effects [39]. It is also important to consider how COVID-19 lockdown conditions may have influenced partner effects. Lockdown conditions may have led to more partner conflict to begin with [17] and/or created substantially more opportunity for partner influence, particularly in the relative absence of other social influences [42]. Ultimately, more research is required to determine the reliability, size, and direction of partner effects in drinking behavior via drinking motives, across samples and contexts.

### 4.3. Limitations

There are several limitations of note. Conclusions on temporality and causality are limited by our cross-sectional design. However, our results largely replicate the Lambe et al. data, which were longitudinal [38]. Additional longitudinal design studies are necessary to strengthen the temporal associations. Lab-based couple studies of elicited romantic conflict [78] would allow for the examination of causal effects. The literature would also benefit from daily diary studies to assess the associations between these constructs at a micro level, i.e., drinking the day of or the day after a romantic conflict, assessing the intensity of conflict at the event level, and assessing motives for specific drinking episodes.

Research has often distinguished between coping with anxiety and coping with depression drinking motives as seen in the five-factor model of drinking motives [49], including in Lambe et al. [38]. Future studies could benefit from using multi-item measures of drinking to cope [49]. The BAMM was selected for this study to minimize participant burden given the other constructs of interest in the larger study, particularly during an already stressful time. But this choice did not allow for the separation of the unique contributions of coping with anxiety and coping with depression motives to the mediation effect observed in the current study. Moreover, our distinguishable dyad analyses only examined mixed-gender couples, excluding gender- and sexuality-diverse relationships. This narrows the generalizability of the gender moderation results to mixed-gendered couples. This limitation is imposed by the nature of the statistical techniques available to date within the APIM framework.

Given numerous known influences on substance use behavior, such as personality (e.g., impulsivity [79], adverse childhood experiences [80,81], genetic risk [82]), it is important to note that this study only examines one possible mechanism influencing alcohol use. Despite well-established distal risk factors, proximal factors (i.e., drinking motives) are often more robust predictors of substance behavior and may be more suitable for intervention [83].

Lastly, the current study conceptualized conflict as a dyadic variable and relied on self-reports. While this is consistent with most studies [47] and allowed for a conceptual replication of Lambe et al. [38], it limited exploration of possible effects of directionality of conflict behaviors (e.g., one partner ignoring, getting angry with, embarrassing the other), or divergent perceptions of conflict behavior, on subsequent coping-motivated drinking behavior. Self-reported conflict behaviors may result in biases [84] in under-reporting socially undesirable behavior such as the perpetration of abusive conflict behaviors [85]. However, there is evidence of conflict predicting subsequent drinking for both directions of conflict behavior (i.e., perpetration and receipt of conflict). Moreover, some research suggests one’s own perception of conflict behavior may be the stronger predictor of drinking [86]. Another notable limitation is that these data relied on retrospective self-reporting. In addition to the possibility of unsystematic error in reporting, retrospective recall may also be susceptible to recall biases [87], particularly for socially undesirable behaviors. However, the novelty of lockdowns in April 2020 may have enhanced individuals’ memories of that time period, as novelty reliably enhances memory of behaviors occurring contiguously with, before, or after a novel event [88].

## 5. Conclusions

There are several implications of these results for research, theory, and practice. Research found increased drinking during the pandemic, which was consistent with the binge drinking rates in the current sample [6]. Our findings suggest couples’ conflict and coping motives are important factors related to greater drinking behavior during the pandemic for both men and women. These results contribute to the existing evidence to encourage more mental health and social supports being allocated for couples cohabitating during the pandemic or who are otherwise socially isolated. This should include increased supports for those experiencing the extremes of a couple’s conflict (i.e., intimate partner violence), which has been shown to be elevated in prevalence during the pandemic [16]. Additionally, these data provide support for the allocation of resources to at-risk couples, given that higher rates of both men and women in our sample were observed to be engaged in risky binge drinking during the pandemic compared to pre-pandemic adult norms [45]. Recent research has evidenced the effectiveness in reducing conflict intensity of brief cognitive reappraisal interventions for couples following conflict during the pandemic [89]; prior research has shown this intervention’s effects on reducing alcohol use [90]. Lastly, the presence of the protective partner effect on women’s drinking behavior via men’s drinking motives has implications: undertreated men’s distress may have downstream consequences for the well-being of their women partners who may compensate to provide emotional support. These results underscore the need to consider the relationship context in individual or couples’ psychological interventions for risky drinking, in addition to examining individual motives for drinking. Overall, this study adds to the growing evidence of the social influence of drinking motives on other individuals’ drinking behavior, including possible protective effects on drinking behavior.

## Figures and Tables

**Figure 1 ijerph-20-06332-f001:**
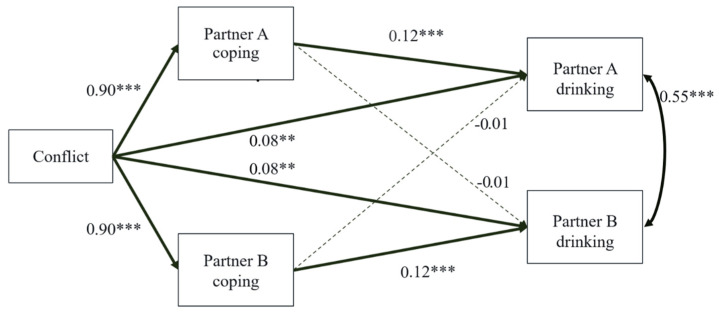
Generalized SEM Model with Indistinguishable Dyads. Values indicate standardized *b* coefficients. *** *p* < 0.001, ** *p* < 0.01.

**Figure 2 ijerph-20-06332-f002:**
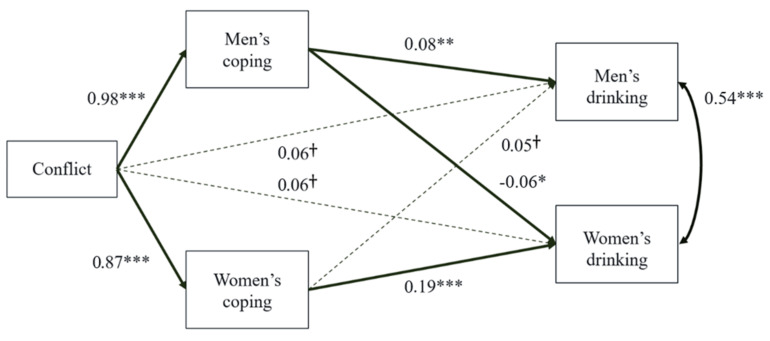
Generalized SEM Model with Distinguishable Dyads. Values indicate standardized *b* coefficients. *** *p* < 0.001, ** *p* < 0.01, * *p* < 0.05 *, † *p* < 0.10.

**Table 1 ijerph-20-06332-t001:** Categorical Demographic Characteristics and Comparisons with Excluded Participants.

Sample Characteristics	*n* (%)	χ2	*p*
Gender		1.04	0.793
Women	340 (48.9)		
Men	355 (51.0)		
Other	1 (0.1)		
Ethnicity		37.36	0.001
White	563 (80.9)		
East Asian	49 (7.0)		
South Asian	24 (3.4)		
Arab/West Asian	6 (0.9)		
Black	11 (1.6)		
Latin American	3 (0.4)		
Indigenous	1 (0.1)		
Multiracial	19 (2.7)		
Other	14 (2.0)		
Unknown	5 (0.7)		
Education level		14.35	0.045 * ^†^
Elementary or some high school	40 (4.9)		
High school graduate	144 (17.5)		
Some college/university	130 (15.8)		
University graduate	371 (45.0)		
Some post-graduate	31 (3.8)		
Post-graduate degree	106 (12.9)		
Prefer not to answer	2 (0.2)		
Income		54.24	0.001 * ^†^
$25,000 or less per year	9 (1.3)		
Between $26,000 and $50,000	62 (8.9)		
Between $51,000 and $75,000	108 (15.5)		
Between $76,000 and $100,000	156 (22.4)		
Between $101,000 and $125,000	114 (16.4)		
Between $126,000 and $150,000	72 (10.3)		
$151,000 or more per year	124 (17.8)		
Prefer not to answer	51 (7.3)		
Employment status		15.46	0.001 * ^†^
Unemployed	292 (42.0)		
Student	5 (0.7)		
Employed part-time	83 (11.9)		
Employed full-time	268 (38.5)		
Relationship status		0.12	0.734
Married or common law	688 (98.9)		
Non-married	8 (1.1)		
Living with children		3.05	0.081
Yes	170 (24.4)		
Relationship composition		3.01	0.222
Mixed-gender	658 (94.5)		
Same-gender	34 (4.9)		
Unknown or other	4 (0.6)		

Demographic characteristics of the sample were compared with excluded couples (i.e., those where one or both partners did not consume alcohol in April 2020). * *p* < 0.05; ^†^ = indicates the given characteristic is significantly higher/larger in the sample used in the present analyses than in excluded couples.

**Table 2 ijerph-20-06332-t002:** Continuous Sample Characteristics and Comparisons with Excluded Participants.

Sample Characteristics	M	SD	Mdn	Range	*t*	*p*
Age	54.55	13.72	57.00	25–87	−0.01	0.990
Relationship length	27.10	14.50	25.20	1.1–62.8	0.01	0.993
Conflict	16.76	12.70	12.00	7–63	−1.59	0.112
Coping motives	2.16	2.88	0.68	0–10	−1.75	0.081
Drinking frequency	12.37	9.85	8.00	1–30	−6.40	<0.001
Peak quantity	4.63	5.57	3.00	1–25.5	−1.12	0.262
Typical quantity	2.79	3.64	2.00	1–25.5	−1.85	0.064

Independent samples *t*-tests were used to compare drinkers (included in main analyses) vs. non-drinkers (couples where one or both members did not drink in April 2020 and were excluded from main analyses) on continuous sample characteristics. A significant and negative *t*-value indicates the given variable was higher/larger in the sample used in the present analyses.

**Table 3 ijerph-20-06332-t003:** Alcohol Use by Gender.

	Men			Women		
Sample Characteristics	M	SD	Binge Drinking (%)	M	SD	Binge Drinking (%)
Drinking frequency	13.66	10.02	n/a	11.03	9.42	n/a
Peak quantity	5.27	5.91	33.3	3.97	5.13	29.8
Typical quantity	3.15	3.77	16.9	2.43	3.46	13.5

Drinking frequency = number of drinking occasions in April 2020; peak quantity = number of drinks consumed on heaviest drinking occasion in April 2020; typical quantity = typical number of drinks consumed per drinking occasion in April 2020. Binge drinking was defined as drinking 5 or more drinks in a single occasion for men and 4 or more drinks in a single occasion for women in April 2020 [45]. n/a = not applicable since binge drinking prevalence can only be calculated from quantity variables. Pre-pandemic, Canadian adults (age 25+) engaged in binge drinking (peak quantity; single occasion in a given month) at rates of 27.1% for men and 19.6% for women [45].

**Table 4 ijerph-20-06332-t004:** Correlation Matrix.

	1	2	3	4	5
1. Conflict	--	0.50 **	0.08 *	0.11 **	0.20 **
2. Drinking to cope		--	0.17 **	0.15 **	0.28 **
3. Drinking frequency			--	0.28 **	0.21 **
4. Peak quantity				--	0.49 **
5. Typical quantity					--
Internal consistency (α)	0.95	0.91	-	-	-

** *p* < 0.01, * *p* < 0.05. Internal consistency in column 2 treats drinking to cope with anxiety and drinking to cope with depression as two items of the single measure of drinking to cope. Drinking frequency = number of drinking occasions in April 2020; peak quantity = number of drinks consumed on heaviest drinking occasion in April 2020; typical quantity = typical number of drinks consumed per drinking occasion in April 2020.

**Table 5 ijerph-20-06332-t005:** Direct and Indirect Effects of Conflict on Drinking in Indistinguishable Dyads.

Effect	Path	*b*	*SE*	*p*	IRR	LCI	UCI
Direct	Conflict → A Drinking	0.08	0.03	0.010	1.08	1.02	1.16
Indirect	Conflict → A Coping → A Drinking	0.11	0.02	<0.001	1.12	1.08	1.16
Indirect	Conflict → P Coping → A Drinking	−0.01	0.02	0.740	0.99	0.96	1.03
Total	Conflict → A Drinking	0.20	0.02	<0.001	1.22	1.15	1.27

IRR = Incident Rate Ratio. LCI and HCI are lower and upper 95% confidence intervals from bias-corrected bootstrapping with 1000 resamples. A = actor; P = partner.

**Table 6 ijerph-20-06332-t006:** Direct and Indirect Effects of Conflict on Men and Women Partners’ Coping and Drinking.

Effect	Path	*b*	*SE*	*p*	IRR	LCI	UCI
Direct	Conflict → M Drinking	0.06	0.03	0.092	1.06	0.99	1.13
Indirect	Conflict → M Coping → M Drinking	0.07	0.02	0.002	1.07	1.03	1.13
Indirect	Conflict → W Coping → M Drinking	0.05	0.03	0.070	1.05	1.00	1.09
Total	Conflict → M Drinking	0.18	0.03	<0.001	1.20	1.14	1.26
Direct	Conflict → W Drinking	0.06	0.04	0.108	1.06	0.99	1.15
Indirect	Conflict → W Coping → W Drinking	0.17	0.03	<0.001	1.19	1.12	1.26
Indirect	Conflict → M Coping → W Drinking	−0.06	0.03	0.033	0.94	0.89	1.00
Total	Conflict → W Drinking	0.17	0.02	<0.001	1.19	1.12	1.27

IRR = Incident Rate Ratio. LCI and HCI are lower and upper 95% confidence intervals from bias-corrected bootstrapping with 1000 resamples. M = men; W = women.

## Data Availability

The data presented in this study are available on reasonable request from the corresponding author.

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
