# Peer review of "Drinking to Cope Mediates the Association between Dyadic Conflict and Drinking Behavior: A Study of Romantic Couples during the COVID-19 Pandemic"

_ijerph, 2023, doi:10.3390/ijerph20146332_

Round 1
Reviewer 1 Report
This is an interesting article that explores the consequences of the interaction between romantic couples on drinking behaviour and seeks to replicate a previous study by Lambe et al. (2015) with a larger sample that includes more older participants.
In Table 1 I found the data reported on Relationship Composition slightly surprising. It suggests that the majority of the sample (94.5%) were same-gender couples. If this is true then the analysis of male to female effects that are reported are only relevant for the remaining 34 cases) seem unlikely. It is more likely, perhaps, that this is a typographical error.
Table 2 which presents the characteristics with excluded participants reports variables prior to an explanation of how they were measured (e.g. Conflict, Coping motives etc.). I think it would be more useful to have this table after the measures are defined so it is easier to interpret the data.
You report the impact of drinking behavior associated with partner’s coping motives (p.10, figure 2) and when discussing (p.12) in section 4.2 you suggest that “Men reported marginally more drinking behavior….” and that this was not statistically significant but that “women reported significantly less drinking….”. However the standardized b coefficients are almost the same size despite one being positive and one negative (0.05 for men and -0.06 for women); if 0.05 is marginal then isn’t -0.06 also marginal? That is, the effect size is small for both genders.
You suggest that ‘emotion work’ may be an explanation for this finding and reflect the gendered nature of emotion work. This may be so but the sources you cite relate to housework and may be different for issues relating to conflict or indeed recreational and leisure activities.
You suggest that one of the contributions of your study is a sample that is not predominantly youth or adult couples but includes ‘middle adulthood’. I was not sure in the analysis if you considered both age and duration of partnership as both direct and indirect effects. That is, older people are likely to have longer relationships and it may be either of these (age or length of partnership) that mediates any interaction effects within the relationship. You may have considered this in you analysis and if so perhaps it is worth making this clearer in your findings. Indeed, you suggest that the difference in the age composition of your sample is a particular strength in the replication of the study by Lambe et al. (2015) so more could have been made of such factors in your analysis and discussion.
Finally, although you acknowledge this in the discussion of the limitations of the study the data is based on retrospective accounts and this is likely to be biased in both systematic and unsystematic ways making a prospective study and the use of other forms of data collection important. I recognise that there is nothing you can do about the nature of your data and that you have noted this limitation and that you suggest that a daily diary study might be a better design I just wanted to emphasize and agree with you on the implications for the design of future studies.
There are some minor typographical errors:
Page 8
3rd paragraph line 6 – “Similarly, an IRR of .95 indicates that every one-unit more inf conflict reported…” should be Similarly, an IRR of .95 indicates that every one-unit more in conflict reported…”
Subtitle 3.2 “Descriptve” should be “ Descriptive”
Reviewer 2 Report
This study examined coping motives as a mediator between dyadic conflict and drinking behavior using actor-partner interdependence models among 348 cohabiting couples during the first lockdown in April 2020. While previous studies have examined the impact of coping motives on drinking behaviors, less is known about coping motives and drinking behavior among couples (in which both partners in the dyad are queried), and how coping motives may mediate the association between dyadic conflict and drinking behavior. This is important because of the influence of romantic partners on each other’s behavior, and the primary importance of the dyadic relationship in general. Overall, this is a very well written paper that addresses a research gap in this area.
The literature review is thorough. The study’s hypotheses are clearly stated.
Materials and Methods
Given the well-documented stressors associated with the pandemic lockdowns, this study sought to recruit cohabiting couples subject to Canadian lockdown ordinances to constitute the sample for this study. By necessity, the study’s sample was restricted to current drinkers (those who drank at least once during April 2020). This is justifiable and the authors present a comparison between the sample and those excluded.
The age distribution of the participants, however, may be a bigger concern. It appears that there was no upper age limit to being in the study. In terms of dyadic conflict, what impact would this have by including older couples? For example, in a conflict-related area, intimate partner violence, these behaviors drop sharply among older (50+ years of age) couples. Most partner violence occurs among younger couples. The authors should address how the age distribution of the sample may have affected the results.
Results
There appears to be an error in Table 1: same gender couples (n=658) and mixed gender couples (n=34). Should these numbers be reversed?
Discussion
It doesn’t appear that exposure to adverse childhood experiences (ACE) or impulsivity were measured in the survey. Given that exposure to ACE is linked to higher levels of impulsivity, and thereafter substance use, do the authors think that the absence of these measurements represents a limitation of the current study?
